Identification and mechanistic insights of cell senescence-related genes in psoriasis

Deng Guiyan 1
Xu Cheng 2 xucheng386@gmail.com
Mo Dunchang 3
1 Department of Dermatology, Nanning Second People’s Hospital , Nanning, Guangxi , China
2 Science and Education Department, Guangxi Zhuang Autonomous Region Jiangbin Hospital , Nanning , China
3 Radiotherapy Department, Nanning Second People’s Hospital , Nanning, GuangXi , China
Nakai Kenta
Electronic publication date: 2025 Jan 14
Publication date: 2025
Volume: 13
Electronic Location ID: e18818
Received 2024 Sep 26; Accepted 2024 Dec 14
Copyright: © 2025 Deng et al.
Copyright year: 2025
Copyright holder: Deng et al.
License: This is an open access article distributed under the terms of the Creative Commons Attribution License, which permits unrestricted use, distribution, reproduction and adaptation in any medium and for any purpose provided that it is properly attributed. For attribution, the original author(s), title, publication source (PeerJ) and either DOI or URL of the article must be cited.
License URL: https://creativecommons.org/licenses/by/4.0/

Keywords: Psoriasis, Cell senescence, DEGs, Machine learning, Immune infiltration

Funding: The authors received no funding for this work.

==============================
Background

Psoriasis is a chronic inflammatory skin disease affecting 2–3% of the global population, characterised by red scaly patches that significantly affect patients’ quality of life. Recent studies have suggested that cell senescence, a state in which cells cease to divide and secrete inflammatory mediators, plays a critical role in various chronic diseases, including psoriasis. However, the involvement and mechanisms of action of senescence-related genes in psoriasis remain unclear.

Methods

This study aimed to identify senescence-related genes associated with psoriasis and explore their molecular mechanisms. RNA sequencing data from psoriasis and control samples were obtained from the GEO database. Differential expression analysis was performed using DESeq2 to identify differentially expressed genes (DEGs). The intersection of DEGs with cell senescence-related genes from the CellAge database was used to identify the candidate genes. Protein-protein interaction networks, Gene Ontology, and Kyoto Encyclopaedia of Genes and Genomes (KEGG) pathway enrichment analyses were conducted to explore the functions and pathways of these genes. Machine learning algorithms, including Least Absolute Shrinkage and Selection Operator (LASSO) regression and Support vector machine-recursive feature elimination (SVE-RFE), were used to select feature genes that were validated by qRT-PCR. Additionally, an immune cell infiltration analysis was performed to understand the roles of these genes in the immune response to psoriasis.

Results

This study identified 4,913 DEGs in psoriasis, of which 46 were related to cell senescence. Machine learning highlighted four key genes, CXCL1, ID4, CCND1, and IRF7, as significant. These genes were associated with immune cell infiltration and validated by qRT-PCR, suggesting their potential as therapeutic targets for psoriasis.

Conclusions

This study identified and validated key senescence-related genes involved in psoriasis, providing insights into their molecular mechanisms and potential therapeutic targets and offering a foundation for developing targeted therapies for psoriasis.

Introduction

Psoriasis (PSO) is a chronic, recurrent, inflammatory skin disease affecting approximately 2–3% of the global population, with a high prevalence in adults (Armstrong & Read, 2020). It is characterised by red scaly patches on the skin that cause significant discomfort and affect the quality of life of patients (Griffiths & Barker, 2007). The pathogenesis of PSO is complex and involves the interplay of genetic, immune, and environmental factors (Petit et al., 2021; Elder et al., 2010; Zhang & Wu, 2018). Despite extensive research, several pathogenic mechanisms remain unclear. Recently, cell senescence has been identified as a critical factor in various chronic and inflammatory diseases. However, its specific role in PSO is not fully understood (Lasry & Ben-Neriah, 2015; Teo et al., 2019; Arra et al., 2022).

Cell senescence is a state induced by various stress factors, in which cells permanently stop dividing and exhibit significant phenotypic changes, including the secretion of numerous inflammatory mediators (Liu, 2022; Bottazzi, Riboli & Mantovani, 2018). These factors, including proinflammatory cytokines, growth factors, and proteases, can disrupt tissue homeostasis and drive chronic inflammation (Davalos et al., 2010; Donato et al., 2015; Yang et al., 2022b). In PSO, such inflammatory mediators may contribute to abnormal skin cell proliferation and inflammation, implicating cell senescence in the disease’s pathogenesis (Lin et al., 2022; Yang et al., 2022a). Studies have shown that cellular senescence is typically elevated in PSO, predominantly affecting specific cell types including keratinocytes, fibroblasts, endothelial cells, and immune cells. Senescence in keratinocytes exacerbates psoriatic inflammation through secretion of the senescence-associated secretory phenotype, whereas fibroblast senescence contributes to aberrant matrix remodelling. Endothelial cell senescence promotes angiogenesis, and immune cell senescence intensifies chronic inflammation (Mercurio et al., 2020; Liu et al., 2023; Zhu et al., 2024; Khmaladze, Nandakumar & Holmdahl, 2015). These senescent cells accumulate in the psoriatic lesions and ultimately drive disease progression (Ghosh & Capell, 2016). Zhu et al. (2024) demonstrated that the topical application of BCL-2 inhibitors can alleviate imiquimod-induced PSO-like skin inflammation by clearing senescent cells, whereas flavonoids were shown to mitigate PSO-like skin lesions by interfering with the process of cellular senescence and suppressing inflammation (Rivera-Yanez et al., 2021). These studies suggest that clearance of senescent cells may represent a novel therapeutic strategy for treating PSO; however, the specific senescence-related genes involved in PSO require further exploration.

This study aimed to identify cell senescence-related genes (CSRGs) associated with PSO and explore their underlying molecular mechanisms by integrating bioinformatics analysis and experimental validation. We obtained RNA sequencing data from PSO and control samples from public databases, used differential expression analysis, protein-protein interaction (PPI) network construction, and functional enrichment analysis to identify potential key genes, and validated these findings using quantitative real-time PCR (qRT-PCR). We also investigated the relationship between these genes and immune cell infiltration to determine their role in PSO. Through this research, we hope to provide new insights into the pathogenesis of PSO and identify potential therapeutic targets. A flow chart depicting the schematic of this study is shown in Fig. 1.

Figure 1 Study design flowchart.

Materials and Methods

Microarray data and cell senescence-related genes

RNA-seq data for the GSE54456 dataset (Wang et al., 2021) (171 samples, 90 PSO and 81 control skin tissues, sequenced using the GPL9052 platform) were downloaded from the GEO database (Edgar, Domrachev & Lash, 2002). The validation dataset (GSE13355 (Zhou et al., 2023b): 122 samples: 58 PSO and 64 control skin tissues, sequenced using the GPL570 platform) was obtained from the GEO database. A total of 279 CSRGs were identified in the CellAge database (Xiang et al., 2022), as described in the Material S1.

Identification of differentially expressed cell senescence-related genes

The DESeq2 package (Love, Huber & Anders, 2014) in R was used to identify differentially expressed genes (DEGs) between PSO and control samples in the GSE54456 dataset. The selection criteria were set to adj p value < 0.05 and |log2FC| > 1. Volcano plots were generated using the ggplot2 package (Wickham, 2009) to visualise the distribution of DEGs, highlighting the top five upregulated and the top ten downregulated genes based on log2FC. Heatmaps of DEGs were created using the ComplexHeatmap package (Gu, Eils & Schlesner, 2016).

Identification of candidate genes

To identify candidate genes related to cell senescence in PSO, the intersection between DEGs and CSRGs was examined.

PPI network construction

To explore the interactions between proteins encoded by the key genes, the STRING database (Szklarczyk et al., 2019) (http://www.string-db.org/) was used to construct a PPI network for the candidate genes.

Chromosomal distribution of genes

To understand the chromosomal distribution of the candidate genes, the OmicCircos package (Hu et al., 2014) was used for visualisation, revealing the genomic structure and functional relationships of these genes, which aids in understanding genetic variations related to PSO.

Gene ontology (GO) and KEGG enrichment analysis

GO enrichment analysis was performed using the GO database to annotate the biological processes (BPs), cellular components (CCs), and molecular functions (MFs) of the genes. Kyoto Encyclopaedia of Genes and Genomes (KEGG) pathway analysis was conducted to annotate metabolic and signalling pathways involving the identified proteins or genes. The top ten enriched pathways were visualised using pathway diagrams.

Machine learning for feature gene selection

Two machine learning algorithms were used to select feature genes from the candidate genes.

LASSO regression

Using the glmnet package (Friedman, Hastie & Tibshirani, 2010), Least Absolute Shrinkage and Selection Operator (LASSO) regression was applied to compress regression coefficients and reduce data dimensions, avoiding multicollinearity and overfitting.

SVM-RFE

Support vector machine-recursive feature elimination (SVM-RFE) was used to iteratively train models, assess feature importance, and remove the least important features, implemented using the caret package (Kuhn, 2008). The intersection of feature genes obtained from both methods was considered a potential candidate gene, and Venn diagrams were plotted using the ggvenn package.

ROC analysis

The discriminatory power of potential candidate genes between the PSO and control samples was evaluated using receiver operating characteristic curve (ROC curves) and area under the curve (AUC values) in the GSE54456 and GSE13355 datasets, plotted using the pROC package (Robin et al., 2011). An AUC value >0.7 was considered indicative of good discriminatory ability.

Expression analysis

Sample collection and RNA extraction

Skin tissue samples were collected from five patients with PSO (experimental group) and five healthy controls (control group). Frozen tissues (50 mg) were processed using TRIzol reagent (Invitrogen, Waltham, MA, USA) for RNA extraction. The tissue was homogenised in 1 mL of TRIzol, incubated on ice for 10 min, followed by the addition of 300 µL chloroform. After vigorous shaking and a 10-min incubation at room temperature, samples were centrifuged at 12,000 g for 15 min at 4 °C to separate RNA in the aqueous phase. The RNA was precipitated using ice-cold isopropanol, washed with 75% ethanol, air-dried, and dissolved in RNase-free water. RNA purity and concentration were assessed using a NanoDrop 2000 spectrophotometer (Thermo Fisher Scientific, Waltham, MA, USA), with an A260/280 ratio between 1.8 and 2.0, indicating purity.

cDNA synthesis

Reverse transcription was performed using the SureScript-First-strand-cDNA-synthesis-kit (Bio-Rad, Hercules, CA, USA). The reaction mixture included 4 µL of 5x Reaction Buffer, 1 µL Primer, 1 µL SweScript RT I Enzyme Mix, 2 µg RNA, and nuclease-free water to a total volume of 20 µL. The reaction conditions were 25 °C for 5 min, 50 °C for 15 min, and 85 °C for 5 s. The cDNA was diluted with RNase-free water (5-20x) before qPCR analysis.

qPCR

It was conducted on a CFX Connect Real-Time PCR Detection System (Bio-Rad, Hercules, CA, USA) using a 20 µL reaction mixture containing 3 µL cDNA, 5 µL 2x Universal Blue SYBR Green qPCR Master Mix, and 1 µL each of forward and reverse primers (10 µM). The cycling conditions were 95 °C for 1 min (initial denaturation), followed by 40 cycles of 95 °C for 20 s (denaturation), 55 °C for 20 s (annealing), and 72 °C for 30 s (extension). Amplification and melting curves were analysed to ensure specificity. The primer sequences were: CXCL1: Forward: AGGCAGGGGAATGTATGTGC, Reverse: GCCCTTTGTTCTAAGCCAGA (amplicon: 203 bp), ID4: Forward: AGCTCCGAAGGGAGTGACTA, Reverse: TCGCTCTGGGTTTTTACGAGG (amplicon: 110 bp), CCND1: Forward: AGCTGTGCATCTACACCGAC, Reverse: GAAATCGTGCGGGTCATTG (amplicon: 93 bp), IRF7: Forward: GCTACAGGCCCTGACCCTCAC, Reverse: GACTCGTCATCTGGAGAGGGT (amplicon: 97 bp), GAPDH (control): Forward: CGAAGGTGGAAGAGTCAACGGATT, Reverse: ATGGGGTGATCATATTCATTG. Primer specificity was verified using Primer-BLAST and melting curve analysis. The PCR efficiency was 96.45%, with a confidence interval of 90–110%, and the coefficient of determination (r2) was 0.9994.

Data analysis

Statistical analyses were conducted using GraphPad Prism software (version 8.0), and the Wilcoxon rank-sum test was used to compare expression differences between the PSO and control groups. All data were presented as mean ± standard deviation (SD), and a p-value of less than 0.05 was considered statistically significant. This study was conducted in accordance with the Declaration of Helsinki and approved by the Ethics Committee of Nanning Second People’s Hospital (Approval No. Y2022112). Written informed consent was obtained from all participants prior to their inclusion in the study. All procedures involving human participants were performed following institutional guidelines to ensure the confidentiality and welfare of the subjects involved.

Nomogram construction

A nomogram was constructed using the RMS package (Harrell, 2015) in R to evaluate the clinical diagnostic value of the key genes. Calibration curves were plotted to assess the predictive accuracy of the nomograms. Hosmer-Lemeshow tests were conducted to evaluate model fit, with p > 0.05, indicating good fit.

Gene set enrichment analysis

Gene set enrichment analysis (GSEA) was performed using the clusterProfiler package (Yu et al., 2012) to explore biological pathways involving key genes. Samples in the GSE54456 dataset were divided into high- and low-expression groups based on the median expression levels of key genes, and pathways were enriched using the MSigDB database (Liberzon et al., 2011) with FDR < 0.25 and p < 0.05.

Functional similarity analysis

The GOSemSim package (Yu et al., 2010) was used to calculate similarity scores to identify functional similarities between the key genes.

Immune infiltration analysis

The CIBERSORT algorithm (Newman et al., 2015) was used to analyse the abundance of 22 immune cell types in both the training (GSE54456) and validation (GSE13355) sets. Wilcoxon rank-sum tests were used to compare the infiltration levels between PSO and control samples, and the results were visualized using ggplot2 (Wickham, 2009).

Disease association analysis

The DisGeNet database (Pinero et al., 2017) (http://www.disgenet.org/) was used to identify other diseases potentially associated with key genes, and Cytoscape was used to visualise the gene-disease co-expression network.

Drug prediction

The DGIdb database (Cotto et al., 2018) (https://www.dgidb.org) was used to identify compounds related to key genes (CXCL1, ID4, CCND1, and IRF7). Drug-gene relationships were visualised in a network.

Regulatory network construction

The multimir package (Ru et al., 2014) was used to identify miRNAs targeting key genes using databases, such as miRDB (Chen & Wang, 2020), TargetScan (Agarwal et al., 2015), and elmmo (Gould et al., 2010). The ENCORI database (Li et al., 2014) was used to predict the lncRNAs that regulate these miRNAs. Cytoscape was used to construct an lncRNA-miRNA-key gene regulatory network.

Results

Differentially expressed genes in PSO

A total of 4,913 DEGs were identified in PSO samples compared to healthy controls. Among these, 1,947 genes were upregulated and 2,966 genes were downregulated. The volcano plot (Fig. 2A) highlights these DEGs, where the red dots represent upregulated genes and the green dots represent downregulated genes. The top five upregulated genes with the highest fold- changes were prominently labelled, as were the top ten downregulated genes. Additionally, a heatmap (Fig. 2B) shows the expression profiles of these top DEGs, illustrating the clear distinction between psoriatic and control samples. The intersection of DEGs and CSRGs yielded 46 intersecting genes, as shown in the Venn diagram in Fig. 3. These intersecting genes were referred to as candidate genes.

Figure 2 Differentially expressed genes (DEGs) in psoriasis.

(A) Volcano plot of DEGs between psoriasis and control samples. Red dots represent upregulated genes, and green dots represent downregulated genes. The top five upregulated and top 10 downregulated genes are labeled. (B) Heatmap of the top DEGs showing the expression proûles in psoriatic and control samples, with red indicating high expression and blue indicating low expression.

Figure 3 Venn diagram of DEGs and cell senescence-related genes (CSRGs).

PPI network of differentially expressed genes

To investigate the PPI network of the identified DEGs, the species was set to “Homo sapiens” with an interaction score threshold of ≥0.7. After removing isolated nodes, 30 candidate genes remained. The resulting TSV file was imported into Cytoscape for network visualisation. As shown in Fig. 4A, the PPI network consisted of 30 nodes and 62 edges, highlighting key interactions among the candidate genes.

Figure 4 Protein-protein interaction (PPI) network and chromosomal distribution of candidate genes.

(A) The protein-protein interaction (PPI) network of 30 candidate genes; (B) the chromosomal distribution of the candidate genes.

Chromosomal distribution of genes

To explore the chromosomal distribution of the candidate genes and determine their roles in genomic structure and function, the OmicCircos package was used to visualise the distribution of these genes on chromosomes. As depicted in Fig. 4B, the Circos plot illustrates the specific locations of candidate genes and other related genes on the chromosomes. The results indicated distinct distribution patterns of candidate genes across different chromosomes, and the connecting lines illustrated the relationships between these genes. These findings suggest potential coregulatory mechanisms or functional collaborations between genes involved in particular BP. Understanding the chromosomal locations and relationships of these genes can further elucidate their association with PSO and aid in the identification of genetic foundations and potential strategies for disease prevention, diagnosis, and treatment.

Biological functions of candidate genes

GO enrichment analysis was performed for 30 candidate genes. Based on a p-value of <0.05, 233 significant GO terms were identified, comprising 195 BPs, 18 CCs, and 20 MFs. As shown in Fig. 5, the most enriched BP terms included mitotic cell cycle phase transition, regulation of the insulin-like growth factor receptor signalling pathway, regulation of epithelial cell differentiation, and replicative senescence. The enriched CC terms included chromosomal regions, condensed chromosomes, and mitotic spindles. Enriched MF terms included insulin-like growth factor I binding, histone kinase activity, and enzyme inhibitor activity.

Figure 5 Gene ontology (GO) enrichment analysis of candidate genes.

(A) The top enriched BP terms; (B) the top enriched CC terms; (C) the top enriched MF terms.

KEGG enrichment analysis

KEGG pathway enrichment analysis was conducted on 30 candidate genes. Based on a p-value < 0.05, 31 KEGG signalling pathways were significantly enriched. As shown in Fig. 6, the top 10 pathways with the smallest p-values included the p53 signalling pathway, cellular senescence, cell cycle, and IL-17 signalling pathway. Detailed pathway diagrams of the top ten enriched pathways are presented in Material S2.

Figure 6 Kyoto Encyclopedia of Genes and Genomes (KEGG) pathway enrichment analysis of candidate genes.

Machine learning for feature gene selection

LASSO regression analysis was performed on 30 candidate genes. As illustrated in Figs. 7A and 7B, 15 genes with nonzero regression coefficients were identified, including AGT, AURKA, CCND1, CDK1, CDKN2A, CXCL1, FOXM1, ID4, IFNG, IGFBP3, IGFBP5, IRF7, MMP9, SERPINE1, and XAF1 (lambda.min = 0.0002895). Further selection using the “caret” R package identified four feature genes: CXCL1, ID4, CCND1, and IRF7 (Fig. 7C).

Figure 7 Machine learning-based selection of feature genes.

(A and B) The LASSO regression analysis, identifying 15 genes with non-zero regression coefficients; (C) the final selection of four feature genes; (D) the Venn diagram shows the intersection of feature genes identified by both LASSO regression and SVM-RFE methods.

Potential candidate genes

The intersection of the feature genes identified by both machine learning methods resulted in four potential candidate genes: CXCL1, ID4, CCND1, and IRF7. A Venn diagram illustrates this intersection (Fig. 7D).

ROC analysis

ROC analysis (Fig. 8) demonstrated that the AUC values for the four potential candidate genes were all greater than 0.7, indicating strong discriminative ability between the PSO and control samples. Thus, CXCL1, ID4, CCND1, and IRF7 were identified as the potential key genes.

Figure 8 ROC analysis of potential candidate genes CXCL1, ID4, CCND1, and IRF7.

The ROC curves illustrate the diagnostic power of these genes in distinguishing psoriasis (PSO) samples from control samples. Each gene’s performance is measured by the area under the curve (AUC), which indicates the gene’s ability to discriminate between the two groups.

Expression analysis

Using the Wilcoxon rank-sum test, the expression levels of the four potential key genes were found to be significantly different (p < 0.05) and consistent in both the training and validation sets (Figs. 9A and 9B). qRT-PCR validation results for the four key genes. Violin plots were generated using ggplot2 (Wickham, 2009) to visualize these differences. The results confirm significant differences in expression levels between PSO patients and controls for CXCL1 (Fig. 9D, p = 0.0318), CCND1 (Fig. 9C, p = 0.0476), and IRF7 (Fig. 9F, p = 0.0296), while ID4 did not show a significant difference in expression between the inflammation group and the control group in the qRT-PCR analysis (Fig. 9E, p = 0.1041), its expression was markedly decreased in patients with PSO based on the gene expression datasets (GSE54456 and GSE13355). This indicates that, despite not reaching statistical significance in the qRT-PCR validation, the overall trend of ID4 expression suggests a downregulation in PSO, which may have potential biological significance in the pathogenesis or progression of the disease. These findings validate the identification of CXCL1, ID4, CCND1, and IRF7 as key genes potentially involved in PSO.

Figure 9 Expression analysis of key genes.

(A) Violin plots of the expression levels of four key genes in the training set; (B) Violin plots of the expression levels in the validation set; (C) qRT-PCR validation showing expression differences in psoriasis and control samples for CCND1; (D) qRT-PCR validation showing expression differences in psoriasis and control samples for CXCL1; (E) qRT-PCR validation showing expression differences in psoriasis and control samples for ID4; (F) qRT-PCR validation showing expression differences in psoriasis and control samples for IRF7. (A, B) ****P < 0.05; (C–F) *P < 0.05; ns P > 0.05.

Construction and evaluation of key gene nomogram

The nomogram (Fig. 10A) demonstrates how the expression levels of CCND1, CXCL1, ID4, and IRF7 could be used to calculate individualised PSO risk scores. Each gene’s expression level was mapped to a corresponding score, and the total score was translated into the PSO risk probability. The calibration curve (Fig. 10B) validated the accuracy of the prediction model, showing that the bias-corrected predicted probabilities (black line) closely matched the ideal situation (dotted line), and the Hosmer-Lemeshow test (p = 0.697) further confirmed good calibration performance.

Figure 10 Nomogram construction and calibration.

(A) Nomogram for individualized PSO risk scores. This nomogram utilizes the expression levels of CCND1, CXCL1, ID4, and IRF7 to calculate personalized psoriasis (PSO) risk scores; (B) calibration curve for prediction model accuracy.

GSEA analysis

Figure 11 shows the enrichment of the four genes (CCND1, CXCL1, ID4, and IRF7) in various KEGG pathways. The curves represent the running enrichment scores, with peak positions indicating the maximum enrichment points of the gene sets in the ranked gene list. The bar graph at the bottom represents gene metrics in the ranked list. CCND1 showed significant enrichment in pathways such as the calcium signalling pathway and cardiac muscle contraction, suggesting its potential role in these processes. CXCL1 was enriched in the chemokine signalling pathway and cytokine receptor interaction, indicating its critical role in immune response. ID4 was significantly enriched in the cytosolic DNA sensing pathway and dilated cardiomyopathy, suggesting that it may play a role in the pathology of cytosolic DNA sensing and dilated cardiomyopathy. IRF7 showed enrichment in immune-related pathways, supporting its role in immune response and inflammation.

Figure 11 GSEA analysis of key genes in various KEGG pathways.

This figure presents the gene set enrichment analysis (GSEA) results for the key genes (CCND1, CXCL1, ID4, and IRF7) within various KEGG pathways.

Friends analysis

The boxplot (Fig. 12) illustrates the functional similarity scores of key genes (CXCL1, IRF7, CCND1, and ID4) in PSO. CXCL1 displayed concentrated distribution with low variability, suggesting consistent functional roles. In contrast, IRF7, CCND1, and ID4 showed broader distributions, indicating significant functional diversity. These results suggest distinct functional roles of different genes in PSO, providing clues for further research.

Figure 12 Functional similarity analysis of key genes.

This figure displays the functional similarity scores among the key genes (CXCL1, ID4, CCND1, and IRF7) involved in psoriasis pathology.

Immune infiltration analysis

Immune infiltration analysis revealed significant differences in immune cell infiltration between the PSO and control samples (Fig. 13). In the training set, 12 immune cell types showed significant differences (p < 0.05) (Figs. 13A and 13C), including activated DCs, Eosinophils, Macrophages (M0, M1, and M2), resting mast cells, Neutrophils, T cells (CD4 memory resting, CD4 naïve, CD8, follicular helper, and regulatory (Tregs)). The validation set revealed 17 immune cell types with significant differences (p < 0.05) (Figs. 13B and 13D), including B cells (memory and naïve), activated DCs, Eosinophils, Macrophages (M0, M1, and M2), mast cells (activated and resting), neutrophils, resting NK cells, Plasma cells, and T cells (CD4 memory activated, CD4 memory resting, CD4 naïve, follicular helper, and Tregs). Despite the differences between datasets, consistent observation of specific immune cells suggests their crucial role in PSO pathology. Further studies on these cellular functions and mechanisms will enhance our understanding of PSO.

Figure 13 Immune infiltration analysis in psoriasis.

(A) Bar plot showing significant differences in immune cell infiltration between PSO and control samples in the training set. (B) Similar bar plot for the validation set. (C) Boxplots of 12 immune cell types with significant differences in the training set. (D) Boxplots of 17 immune cell types in the validation set. *P < 0.05, **P < 0.01, ***P < 0.001, ****P < 0.0001, ns P > 0.05.

Correlation between key genes and differential immune cells

In the training set, DCs (activated and resting) and resting mast cells showed high correlations with key genes (Fig. 14A). In the validation set, activated DCs, neutrophils, T cells (CD4 memory activated), and activated mast cells were highly correlated with the key genes (Fig. 14B). Both datasets suggest that activated DCs and resting mast cells play significant roles in the PSO immune response. Additional findings in the validation set (neutrophils and activated CD4 + memory T cells) highlight their potential roles in PSO immune pathology, warranting further investigation.

Figure 14 Correlation between key genes and differential immune cells.

(A) The correlation between key genes and immune cells in the training dataset; (B) the correlation between key genes and immune cells in the validation dataset.

Disease association analysis of key genes

The gene-disease co-expression network visualised in Cytoscape (Fig. 15A) included 41 nodes (four key genes, 37 related diseases) and 76 edges, illustrating the connections between these genes and various diseases.

Figure 15 Gene-disease co-expression network, drug prediction, and molecular regulatory network.

(A) Gene-disease co-expression network: the connections between key genes (CXCL1, ID4, CCND1, and IRF7) and various diseases are depicted, highlighting their broader relevance in pathological conditions beyond psoriasis. (B) Drug-gene interaction network: shows the interactions between drugs and key genes, suggesting potential therapeutic targets in psoriasis. (C) lncRNA-miRNA-gene regulatory network: illustrates the regulatory relationships among long non-coding RNAs (lncRNAs), microRNAs (miRNAs), and key genes, revealing the intricate regulatory mechanisms in psoriasis.

Drug prediction

As shown in Fig. 15B, the drug-gene network included 12 nodes (two genes, 10 drugs) and 10 edges. Specifically, CCND1 is associated with six drugs (abemaciclib, bexarotene, lapatinib, palbociclib, ribociclib, tamoxifen), which are primarily used for cancer treatment but may also influence PSO through CCND1 modulation. CXCL1 is associated with four drugs (pioglitazone hydrochloride, zafirlukast, troglitazone, and uracil), indicating its role in inflammation and other disease treatments, and potentially mitigating PSO symptoms by inhibiting CXCL1-mediated inflammation. This drug-gene network underscores the potential of CCND1 and CXCL1 as therapeutic targets for PSO.

Construction of molecular regulatory network

An lncRNA-miRNA-key gene regulatory network was constructed using Cytoscape, which revealed complex interactions among the key genes (CCND1 and ID4), miRNAs, and lncRNAs (Fig. 15C). The network included 47 nodes (two key genes, 22 miRNAs, and 23 lncRNAs) and 143 edges. CCND1 interacted with multiple miRNAs (hsa-miR-195-5p, hsa-miR-93-5p, and hsa- miR-130a-5p), whereas ID4 interacted with miRNAs (hsa-miR-424-5p, hsa-miR-497-5p, and hsa-miR-19a-3p), highlighting their central roles. LncRNAs such as NEAT1, TUG1, and MALAT1 indirectly regulate key genes through miRNA interactions, emphasising their regulatory importance. This complex network revealed the intricate regulatory mechanisms of CCND1 and ID4, offering insights into their roles in PSO. Further research on these interactions may uncover new therapeutic targets for PSO treatment.

Discussion

In this study, we identified the key CSRGs associated with PSO. We identified 46 candidate genes through differential expression and cell-senescence gene intersection analyses. Machine learning highlighted four potential key genes, CXCL1, ID4, CCND1, and IRF7, which showed strong discriminatory power and significant differences in expression, as validated by qRT-PCR. GO and pathway analyses revealed the involvement of these genes in critical BPs, including immune responses and cell cycle regulation. These findings provide new insights into the molecular mechanisms underlying PSO, and suggest potential targets for therapeutic intervention.

The four key genes identified in this study play crucial roles in the pathogenesis of PSO through their involvement in cellular senescence and immune response mechanisms. CXCL1 is a chemokine known for its role in the inflammatory response, which is consistent with previous studies highlighting its elevated expression in psoriatic lesions (Zou et al., 2023; Zou, Kong & Sang, 2022; Jiang, Hinchliffe & Wu, 2015). ID4, a transcriptional regulator, has been less studied in PSO; however, our findings suggest its involvement in the molecular pathways of the disease, potentially through the regulation of cellular differentiation and proliferation. CCND1, a cell cycle regulator, and IRF7, an interferon regulatory factor, were also implicated, aligning with previous research indicating their roles in cell proliferation and immune response in PSO (Liu, Schwam & Chen, 2022; Mohd, Azlan & Mohd, 2022; Li et al., 2022). Based on our experimental results, we propose the following mechanisms for the roles of these four key genes in PSO: downregulation of ID4 and CCND1 inhibits cell proliferation and disrupts skin repair processes, leading to impaired epidermal barrier function. In contrast, the upregulation of CXCL1 and IRF7 amplifies local immune responses, promotes cellular senescence, exacerbates skin damage, and forms a vicious cycle that ultimately contributes to the chronicity of PSO. These gene regulatory mechanisms may serve as novel targets for future therapeutic strategies. For example, modulating the expression of ID4 and CCND1 may enhance skin repair and cell proliferation, whereas suppressing the excessive expression of CXCL1 and IRF7 may mitigate inflammatory responses and accumulation of senescent cells, thereby alleviating the clinical symptoms of PSO.

Our GO and KEGG pathway analyses further elucidated the biological functions and pathways in which these genes were involved, such as p53 signalling and cellular senescence pathways, highlighting their potential roles in the progression of PSO (Moorchung et al., 2015; Yokota et al., 2022). Specifically, the enrichment of BPs such as the mitotic cell cycle and immune response components underscores the intricate interplay between cell senescence and PSO pathogenesis, providing novel insights into potential therapeutic targets (Wang & Lai, 2024; Zhou et al., 2024).

Our study found significant correlations between key genes (CXCL1, ID4, CCND1, and IRF7) and specific immune cell types, such as activated DCs and resting mast cells, which are known to play critical roles in the immunopathogenesis of PSO (Rendon & Schakel, 2019; Petit et al., 2021; Zhou et al., 2023a). Activated DCs, which are pivotal for initiating and sustaining the inflammatory response in PSO, showed a strong correlation with CXCL1 and IRF7 expression. This is consistent with previous studies highlighting the role of DCsin PSO through cytokine production and T cell activation (Zhou et al., 2023b; Ozawa & Aiba, 2004). Additionally, resting mast cells, which can transition to an activated state and contribute to inflammation and tissue remodelling, were significantly associated with CCND1 and ID4 expression. This finding aligns with studies that implicate mast cells in the chronic inflammatory environment of psoriatic lesions (Liu, Schwam & Chen, 2022; Mohd, Azlan & Mohd, 2022; Zhou et al., 2023b). GO and KEGG analyses provided further insights, revealing that enriched BPs, CCs, and MFs were closely linked to immune responses and cell cycle regulation. Specifically, the p53 signalling pathway and cellular senescence pathway, both significantly enriched, underscore their potential roles in PSO pathology by modulating cell proliferation, apoptosis, and senescence (Liu, 2022; Zou et al., 2023). These results highlight the intricate interactions between immune cells and senescence-related genes, providing a deeper understanding of the molecular mechanisms underlying PSO and suggesting new avenues for therapeutic interventions.

The construction of the gene-disease co-expression network and—the gene interaction network in this study provides significant insights into potential therapeutic targets for PSO. The gene-disease network revealed associations between key genes (CXCL1, ID4, CCND1, and IRF7) and various diseases, underscoring their broader relevance in pathological conditions beyond PSO. The drug-gene network highlighted several drugs, such as abemaciclib, pioglitazone, and tamoxifen, which are associated with CCND1 and CXCL1 and hold the potential for repurposing in PSO treatment because of their roles in modulating inflammation and cell proliferation (Yang et al., 2020; Lajevardi et al., 2015; Clement et al., 2021). Furthermore, the construction of the lncRNA-miRNA-key gene regulatory network elucidated the complex regulatory mechanisms involving key genes. This network identified core nodes, such as miR-195-5p, and lncRNAs, such as NEAT1, that regulate the expression of key genes, thereby influencing the pathogenesis of PSO. Understanding these interactions provides a more comprehensive view of the underlying molecular mechanisms and highlights potential targets for novel therapeutic interventions. Identification of these regulatory nodes is crucial because they may serve as biomarkers or therapeutic targets, offering new avenues for personalised treatment strategies for PSO (Shen et al., 2024; Akhlaq et al., 2023). These findings emphasise the importance of integrating multi-layered genetic and epigenetic data to uncover the underlying molecular mechanisms and therapeutic targets of PSO.

Although our study provides significant insights into the molecular mechanisms of cellular senescence-related genes in PSO, it has some limitations. One major limitation is the reliance on publicly available datasets, which may introduce selection bias and limit the generalisability of our findings. Additionally, the sample size may not have fully captured the genetic diversity and environmental influences of the broader population with PSO. Bioinformatics analyses depend on the quality and completeness of the databases and algorithms used. Furthermore, the experimental validation was limited to qRT-PCR, necessitating extensive functional studies to confirm the roles of the identified key genes.

Conclusions

In conclusion, this study enhances our understanding of the association between CSRGs and PSO. By integrating bioinformatics and experimental validation, we identified key genes and their roles in this disease. This research underscores the importance of genetic and molecular insights for developing targeted therapies. Future studies should focus on validating these results in larger, more diverse populations and conducting in-depth functional analyses. Exploring epigenetic influences and employing multi-omics approaches will provide a more comprehensive understanding of the mechanisms underlying PSO. Longitudinal studies will help understand the dynamic changes in gene expression and immune profiles over time and in response to treatments and will pave the way for personalised medicine in patients with PSO, aiming for targeted, effective therapeutic strategies based on individual molecular profiles.

Supplemental Information

Supplemental Information 1 MIQE checklist.

Supplemental Information 2 Cell Senescence-related Genes.

Supplemental Information 3 Detailed Top 10 KEGG Pathway Analysis Results.

Supplemental Information 4 PCR raw data.

Supplemental Information 5 Raw data.

Additional Information and Declarations

Competing Interests

Author Contributions

Human Ethics

Data Availability

The authors declare that they have no competing interests.

Guiyan Deng performed the experiments, analyzed the data, authored or reviewed drafts of the article, and approved the final draft.

Cheng Xu conceived and designed the experiments, prepared figures and/or tables, authored or reviewed drafts of the article, and approved the final draft.

Dunchang Mo analyzed the data, prepared figures and/or tables, and approved the final draft.

The following information was supplied relating to ethical approvals (i.e., approving body and any reference numbers):

The Medical Ethics Committee of the Second People’s Hospital of Nanning approved this study (Approval No: Y2022112).

The following information was supplied regarding data availability:

The RNA-seq data are available at GEO: GSE54456. The validation set data were also available at GEO: GSE13355. The raw data are available in the Supplemental Files.

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
