# Peer review of "Identification and mechanistic insights of cell senescence-related genes in psoriasis"

_PeerJ, doi:10.7717/peerj.18818_

## Round 0.1 · original submission · Major Revisions

Four experts in the field have reviewed your manuscript. Three of them recommend minor revision while Reviewer 4 recommends a major revision. Please read their comments carefully, and revise the manuscript. Notably, Reviewer 4 is concerned about the novelty of this work compared with past works. Reviewer 1's comment also looks fundamental.

Reviewer 1 ·

Basic reporting

Please Provide the evidence that the disturbance of cell senescence itself may generate psoriasis; up- or down regulation of cell senescence may improve psoriasis in mice models or humans?
Though some cytokines like IL17 may alter cell senescence, the cytokines themselves can be targets of psoriasis therapy and alteration of cell senescence may be one of the effects of these cytokines and may not be therapeutic targets by itself.
How cell senescence is altered in psoriasis should be described in introduction; reduced or enhanced? In who cell types?
How the four target cell senescence genes generate psoriasis lesions via up or downregulation of cell senescence should be described even if speculatively.

Experimental design

Mostly well designed.

Validity of the findings

Valid

·

Basic reporting

English must be improved

Experimental design

The research is original and relevant and is within the objectives and scope of the journal.
The research contributes to the knowledge of psoriasis and was carried out with high technical and ethical standards.
The methods were very detailed and provided enough information to be replicated.

Validity of the findings

The conclusions are related to the original research question and the results found.

Additional comments

N/A

·

Basic reporting

The manuscript is interesting with novel idea and good English language, please find below some comments to improve the current manuscript:

In subjects and methods section:

In number 9 “Gene expression analysis”: The authors started this section stating “Wilcoxon rank-sum tests were used to identify significant differences in gene expression between PSO and control samples in both training and validation sets. Violin plots were generated using ggplot2 to visualize these differences”. This paragraph is not related to the subtitle gene expression for validating the results. Please revise the proper location for this paragraph.

In number 9 “Gene expression analysis”: The authors repeated twice the description for RNA extraction method. Please revise this section as it needs many modifications.

How the authors selected the sample size (5 psoriasis tissue specimen and 5 normal skin as controls) for the validation group?

In results section:
The figures caption needs more illustration and adding what the abbreviations included stands for and also add the reference for the software or tool used to generate this figure.

Experimental design

In subjects and methods section:

In number 9 “Gene expression analysis”: The authors started this section stating “Wilcoxon rank-sum tests were used to identify significant differences in gene expression between PSO and control samples in both training and validation sets. Violin plots were generated using ggplot2 to visualize these differences”. This paragraph is not related to the subtitle gene expression for validating the results. Please revise the proper location for this paragraph.

In number 9 “Gene expression analysis”: The authors repeated twice the description for RNA extraction method. Please revise this section as it needs many modifications.

How the authors selected the sample size (5 psoriasis tissue specimen and 5 normal skin as controls) for the validation group?

Validity of the findings

No comment

Reviewer 4 ·

Basic reporting

no comment

Experimental design

no comment

Validity of the findings

no comment

Additional comments

The manuscript describes the investigation of the role of senescence-related genes in psoriasis by means of bioinformatics analysis. Some data derived from those analyses were further verified in skin tissues from psoriasis patients and healthy individuals.
However, several papers are available on cellular senescence in psoriasis; even in these papers authors have discussed the roles of senescent CD4+ T cells in developing psoriasis. I am wondering what authors are gaining from this study.
In line 81 “Differentially Expressed Autophagy-related Genes” should be defined. In line 246 “Biological Functions of Differentially Expressed Genes” should be “Biological Functions of candidate genes”.

---

## Round 0.2 · accepted · Accept

Since all three original reviewers are now satisfied with your revision, I am happy to recommend its acceptance to the section editor. Congratulations!

Reviewer 1 ·

Basic reporting

Most of the decription is well revised.

Experimental design

wel done

Validity of the findings

Valid

Additional comments

No other comments

·

Basic reporting

The authors have responded to the reviewer's comment and it is convenient for me in the current form.

Experimental design

The authors have responded to the reviewer's comment and it is convenient for me in the current form.

Validity of the findings

The authors have responded to the reviewer's comment and it is convenient for me in the current form.

Reviewer 4 ·

Basic reporting

no comment

Experimental design

no comment

Validity of the findings

no comment

Additional comments

no comment